# Meta-Learning without Data via Wasserstein Distributionally-Robust Model Fusion

**Zhenyi Wang**[1]  **Xiaoyang Wang**[2]  **Li Shen**[3]  **Qiuling Suo**[2]  **Kaiqiang Song**[2]  **Dong Yu**[2]  **Yan Shen**[1]  **Mingchen Gao**[1]

[1]Department of Computer Science and Engineering, State University of New York at Buffalo, NY, USA
[2]Tencent AI Lab, Seattle, WA, USA
[3]JD Explore Academy, Beijing, China

## Abstract

Existing meta-learning works assume that each task has available training and testing data. However, we can only use many available pre-trained models without accessing their training data in practice. We often need a single model to solve different tasks simultaneously as this is much more convenient to deploy the models. Our work aims to meta-learn a model initialization from these pre-trained models without using corresponding training data. We name this challenging problem setting Data-Free Learning To Learn (DFL2L). We propose a distributionally robust optimization (DRO) framework to learn a black-box model to fuse and compress all the pre-trained models into a single network to address this problem. The proposed DRO framework diversifies the learned task embedding associated with each pre-trained model to cover the diversity in the underlying training task distributions, encouraging good generalization to unseen new tasks. We sample a meta-initialization from the black-box network during meta-testing for fast adaptation to unseen new tasks. Extensive experiments on offline and online DFL2L settings and several real image datasets demonstrate the effectiveness of the proposed methods.

## 1 INTRODUCTION

The goal of meta-learning is to learn prior knowledge from many similar tasks such that the learned knowledge can be fast adapted to new unseen tasks. Existing meta-learning methods assume that each task has available training and testing data. However, in many real scenarios, each task only has a pre-trained model, and the task-specific data may not be available after training due to privacy issues. For example, many pre-trained network models are available on GitHub without sharing training data. However, sometimes, we need a single model to cover the multiple task knowledge. Furthermore, even if we sometimes can access the corresponding training datasets, it is prohibitive to retrain a model using multiple datasets from scratch due to the enormous computing resources needed to train the large models, such as BERT [Devlin et al., 2018] and GPT3 [Brown et al., 2020]. Thus, meta-learning from those pre-trained models to learn an initialization becomes the central problem so that the fused model can be fast adapted to the new unseen task with only a few labeled data. We name this challenging problem setup as Data-Free Learning To Learn (DFL2L) to reflect that there is no available private data for each task to centralize the meta-learning process, or retraining the model from scratch is too costly.

According to whether pre-trained models are all collected in advance or not, we categorize the DFL2L learning setting into offline and online learning scenarios. Offline DFL2L considers the setting that a fixed number of pre-trained models are available together before performing meta-training. By contrast, online DFL2L assumes that pre-trained models arrive sequentially during meta-training. We update the aggregated network to include the newly arrived pre-trained model. Figure 1(a) and 1(b) show the learning paradigm of offline and online DFL2L problems. For offline DFL2L, all the pre-trained models are collected before meta-training and can be trained for multiple rounds. For online DFL2L, pre-trained models arrive one by one, and the previous models are no longer available when training on a new model. The online setting is similar to the online meta-learning [Finn et al., 2019a], where tasks arrive in sequential order. Thus, online DFL2L is more challenging than offline learning because when learning on the newly arrived pre-trained model, previous ones are not available to learn again.

We propose a black-box distributionally robust optimization (DRO) framework to fuse different pre-trained models into a single network to address this problem. Specifically, we first use a black-box neural network to predict the model parameters for each pre-trained model by using the task

*Accepted for the 38th Conference on Uncertainty in Artificial Intelligence*  (UAI 2022).

embedding as input and maximizing the likelihood of pre-trained model parameters. Next, we cast DFL2L as a bi-level optimization from a meta-learning perspective. Moreover, intuitively, the more diverse the task embedding, the better generalization for the new unseen tasks. To achieve this goal, we further encourage the learned task embedding to be as diverse as possible with DRO by perturbing the task embedding within a Wasserstein ball and optimizing the model performance under the worst-case task embedding. We use the ambiguity set of task embedding with distribution perturbation (Wasserstein ball) to approximate the underlying task distribution uncertainty. Since this ambiguity set contains an infinite number of distributions, it can encourage diverse task distributions that would cover more tasks and improve generalization. The DRO guarantees the model fusion performance under the worst-case scenarios. To solve this optimization, we convert it into an unconstrained optimization by Lagrange multipliers. During meta-testing, the learned meta initialization serves as the model initialization for the unseen testing tasks. It can be effectively adapted to new tasks with only a few labeled data. Our proposed methods can be applied to both online and offline DFL2L settings. Specifically, for the offline DFL2L (see Figure 1(a)) , we apply the proposed method to this setting by training the model with multiple epochs. On the other hand, for the online DFL2L (see Figure 1(b)) , we fuse the model sequentially without revising previously seen pre-trained models.

To evaluate the effectiveness of the proposed method, we construct several benchmarks with different types of pre-trained models. We perform extensive experiments on offline and online pre-trained model fusions and achieve significant improvement compared to state-of-the-art (SOTA) baselines. For our proposed benchmarks with CIFAR-FS and Mini-ImageNet pre-trained models, our method improves over baselines in the range of 2% to 7%, demonstrating the effectiveness of the proposed approach.

To the end, we summarize our contributions as three-fold:

- We propose a new meta-learning paradigm, i.e., data-free learning to learn, dubbed DFL2L, whose goal is to meta-learn a model initialization from these (sequentially arrived) pre-trained models and use it to initialize new unseen tasks. Correspondingly, we construct a set of new and challenging benchmarks.

- We propose a black-box distributionally robust meta-learning framework for learning the meta initialization, which fuses several pre-trained models into a single model without requiring additional training data.

- We apply the proposed method to both offline and online DFL2L settings. Experiments on various benchmarks demonstrate the value of the proposed DFL2L setting and verify the effectiveness of the proposed method on both offline and online DFL2L settings.

## 2 RELATED WORKS

We summarize the most related work to our proposed DFL2L settings and proposed solutions.

### 2.1 OFFLINE META LEARNING

Meta-learning [Schmidhuber, 1987, Naik and Mammone, 1992, Bengio et al., 1997] focuses on extracting common knowledge from many related tasks. Most existing works [Vinyals et al., 2016, Finn et al., 2017, Snell et al., 2017, Ravi and Beatson, 2019, Rajeswaran et al., 2019, Finn et al., 2019b, Raghu et al., 2020, Wang et al., 2020b, Zhou et al., 2021, Bohdal et al., 2021, Rothfuss et al., 2021, CHEN et al., 2021, Bronskill et al., 2021, Sun et al., 2021] focus on the offline setting, where all the training tasks are available together upfront. These works assume that each training task consists of labeled training and testing data. Completely different from those works, our offline DFL2L setting considers the data-free learning-to-learn scenario: we only have a pre-trained model for each task. Thus, existing meta-learning approaches are not applicable in DFL2L.

### 2.2 ONLINE META LEARNING

Online meta-learning (OML) [Finn et al., 2019a, Denevi et al., 2019, Yao et al., 2020, Babu et al., 2021, Wang et al., 2021, 2022] extends meta-learning to the online setting, where tasks sequentially arrives. *Continual-MAML* [Caccia et al., 2020] addresses the problem of fast online adaptation to new tasks while maintaining acquired knowledge on previously learned tasks; MOCA [Harrison et al., 2020] utilizes context data from previous tasks to improve future sequential prediction without knowing when the latent task changes. MOML [Acar et al., 2021] is a memory-efficient version of OML. However, those settings are entirely different from our work, which does not need task-specific data. The proposed online DFL2L setting can be viewed as a more challenging data-free OML.

### 2.3 MODEL FUSION

Model fusion [Yurochkin et al., 2019b,a, Wang et al., 2020a, Lam et al., 2021] is a recent emerging research area to fuse multiple pre-trained models without using their task data to do retraining. Many works are in the context of federated learning McMahan et al. [2017], Shamsian et al. [2021]. [Yurochkin et al., 2019b,a] proposes a Bayesian non-parametric framework that considers the neuron matching in a probabilistic manner but only works with simple architecture, e.g., a fully connected network. Federated Matched Averaging (FedMA) [Wang et al., 2020a] develops a layer-wise weighted averaging method to fuse multiple networks. Similar to FedMA, [Singh and Jaggi, 2020] utilizes opti-

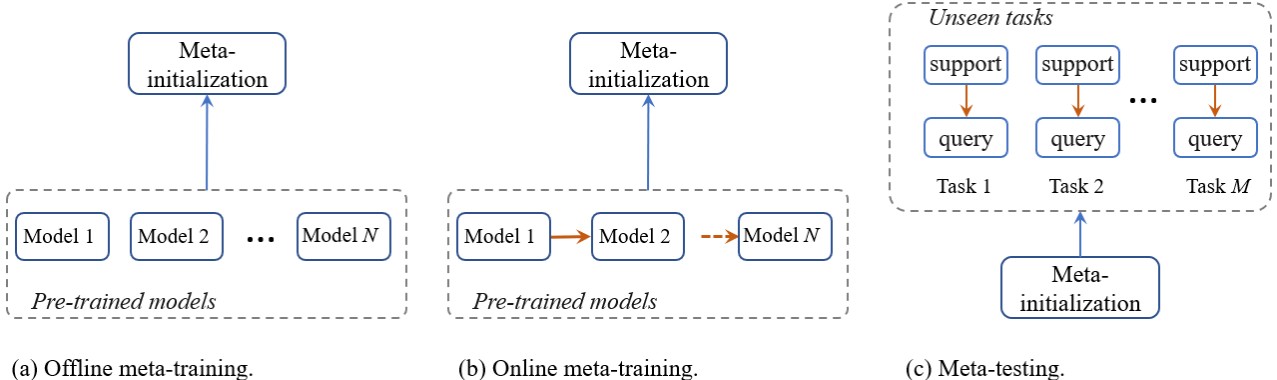

(a) Offline meta-training.    (b) Online meta-training.    (c) Meta-testing.

Figure 1: The data-free learning to learn problem. (a) is the offline meta-learning paradigm, (b) is the online extension, and (c) is meta-testing. In (a), meta-training is performed on a set of pre-trained models, while in (b), it is sequentially trained on one pre-trained model after another. In (c), meta-testing is performed on a collection of unseen tasks for both the offline and online setups, and each test task has few labeled examples for gradient steps of adaptation.

mal transport to align neurons across models in a layer-wise fashion. MFGP [Lam et al., 2021] uses the Gaussian process (GP) for model fusion but only works with fully connected networks. These methods assume all the pre-trained models solve the *same task* and they fuse the models in an offline way. In contrast, we focus on the more general and challenging setting where each model solves a *different* task, and we can fuse the models offline or online via a meta-learning perspective. Thus, the problem setup of DFL2L significantly expands the application scope of model fusion in real scenarios.

## 2.4 DISTRIBUTIONALLY ROBUST OPTIMIZATION

Distributionally Robust Optimization (DRO) [Rahimian and Mehrotra, 2019, Sinha et al., 2018] is an effective optimization framework for handling decision-making under uncertainty. DRO aims to optimize model performance under the worst-case perturbed distribution. We can characterize the underlying uncertainty in various ways, including KL-divergence, Wasserstein ball, etc. DRO has been applied to many machine learning problems, including federated learning [Deng et al., 2021], group shift [Sagawa et al., 2020] and reinforcement learning [Smirnova et al., 2019]. This paper adopts the Wasserstein ball to characterize the task embedding uncertainty since a fixed number of pre-trained models cannot capture the underlying complex task distributions. This ambiguity set allows covering the novel task embedding outside the task embeddings of pre-trained models. To our best knowledge, we are the first to develop a DRO framework for the DFL2L setting to learn a meta-initialization.

## 3 PROBLEM DEFINITION

In this section, we clarify the definitions and scenarios for offline and online DFL2L problem settings. We also discuss the common meta-testing procedure for both offline and online DFL2L.

### 3.1 OFFLINE DFL2L SETUP

Given a collection of $N$ tasks consisting of $\mathcal{C} = (\mathcal{T}_1, \boldsymbol{\theta}_1), \cdots, (\mathcal{T}_N, \boldsymbol{\theta}_N)$, where $\mathcal{T}_i$ is the task identifier and $\boldsymbol{\theta}_i$ are the pre-trained model parameters. We denote the function represented by the network with parameters $\boldsymbol{\theta}_i$ by $H_{\boldsymbol{\theta}_i}$. We assume that all the tasks use the same architecture, the more general and difficult cases that each task uses different architecture is left as future work. The goal is to meta-learn from $\mathcal{C}$ to learn a model initialization $\boldsymbol{\theta}_{init}$ that can be fast adapted to unseen tasks with only a few labeled examples. Figure 1(a) illustrates the offline learning setting.

### 3.2 ONLINE DFL2L SETUP

The offline DFL2L setting assumes all the pre-trained models are available during meta training. In contrast, the pre-trained model sequentially arrives in online DFL2L setting, i.e., $\mathcal{C} = (\mathcal{T}_1, \boldsymbol{\theta}_1), \cdots, (\mathcal{T}_i, \boldsymbol{\theta}_i), \cdots, (\mathcal{T}_N, \boldsymbol{\theta}_N)$, where $\mathcal{T}_i$ is the task identifier and $\boldsymbol{\theta}_i$ are the pre-trained model parameters received at time $i$. Similar to the offline scenario, we assume that all the tasks use the same architecture. The goal is to sequentially meta-learn an initialization $\boldsymbol{\theta}_{init}$ so that it can be fast adapted to unseen tasks with only a few labeled examples. The online setting is shown in Figure 1(b).

## 3.3 META TESTING

During testing, for both offline and online DFL2L, suppose another $M$ unseen tasks arrive together, and each task has its own few labeled data for each class. The goal is to adapt to these labeled data so that the model can perform well on the testing data of each new task. The final accuracy is the average accuracy for these unseen tasks. The meta-testing step is shown in Figure 1(c).

## 4 METHODS

To address the DFL2L problem, we propose a black-box DRO framework to fuse different pre-trained models into a single one. We propose the general framework in Section 4.1, and our improved DRO framework in Section 4.2, including an extension to the offline and online settings. Figure 2 illustrates the overall framework.

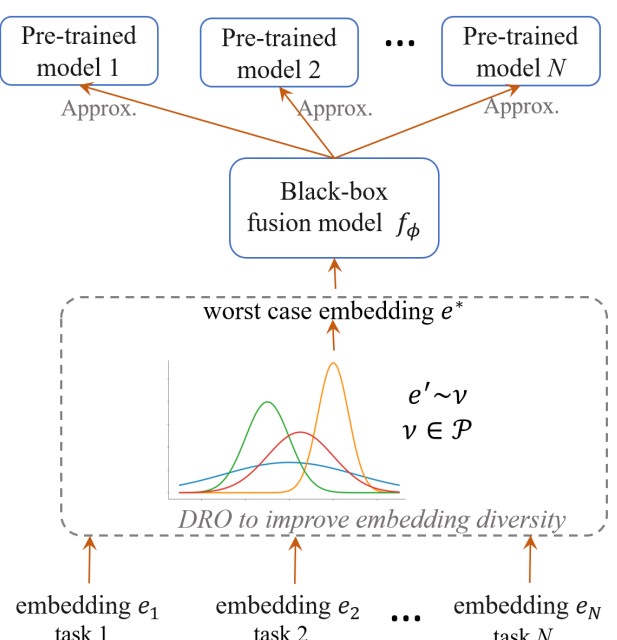

Figure 2: DRO-based black-box model fusion framework. Each training task consists of a task identifier and a pre-trained model. We first embed the task identifier as a vector $e_i$. We adopt distributionally robust optimization (DRO) to approximate diverse task embeddings to capture the underlying uncertainty over task distributions. Specifically, we optimize the worst-case embedding distribution to ensure good performance over all possible perturbed embedding distributions $\nu$ within a Wasserstein ball. We fed the optimized embedding $e_i^*$ into the black-box fusion model $f_\phi$ to obtain meta-initialization, which is encouraged to approximate each pre-trained model as close as possible.

## 4.1 LEARNING OBJECTIVE FOR DFL2L

For each task identity $\mathcal{T}_i$, we first embed the task identifier as the corresponding task embedding $e_i$. We then use a black-box network to fuse different pre-trained models. Suppose the parameters of the black-box model are $\phi$. We use the black-box function $f_\phi(e_i)$ for fitting the task $\mathcal{T}_i$ parameters with task embedding $e_i$ as input. The goal is to optimize the following objective:

$$\max_\phi \sum_{i=1}^{i=N} \log P(\boldsymbol{\theta}_i | f_\phi(e_i)). \quad (1)$$

The likelihood function $P(\boldsymbol{\theta}_i | f_\phi(e_i))$ measures how far away of the black-box network prediction from the true pre-trained task parameters. The likelihood function of $\boldsymbol{\theta}_i$ follows the Gaussian likelihood function:

$$P(\boldsymbol{\theta}_i | f_\phi(e_i)) = exp(-\frac{||f_\phi(e_i) - \boldsymbol{\theta}_i||^2}{\sigma^2}), \quad (2)$$

Where $\sigma$ is a standard deviation constant. This optimization objective is to fuse the knowledge of all the pre-trained models into a single black-box network. The fused network is expected to maintain the maximal information from all the pre-trained models. Although some existing works on federated learning [Wang et al., 2020a] point out that it is beneficial to match the neurons layer-wise due to the permutation invariance of the neural network parameters. This property means that there are many equivalent networks for a given neural network. They only differ in model parameters by permuting the original model parameters. However, these matching methods are not suitable for DFL2L since each model is to solve a different task; there is no correspondence relation for the parameters in a single layer among different pre-trained models. In our preliminary experiments, we found that these matching techniques do not help the training of the black-box model.

To make the model fused by the black-box fusion network generalizable on unseen tasks, the pre-trained models are divided into meta-training collection $\mathcal{S}$ and meta-validation (unseen) collection $\mathcal{Q}$, i.e., $\mathcal{S} \cup \mathcal{Q} = \mathcal{C}$ and $\mathcal{S} \cap \mathcal{Q} = \emptyset$. We focus on solving the following bi-level optimization problem that directly optimizes the generalization on unseen tasks:

$$\phi_{meta} = \arg\max_\phi [\mathcal{F}(\phi) = \mathbb{E}_{\mathcal{T}_j \in \mathcal{Q}} \log P(\boldsymbol{\theta}_j | f_{Alg^*(\phi)}(e_j))]$$
$$Alg^*(\phi) = \arg\max_\omega \mathbb{E}_{\mathcal{T}_i \in \mathcal{S}} \log P(\boldsymbol{\theta}_i | f_\omega(e_i)), \quad (3)$$

where $Alg^*(\phi) = \phi - \nabla_\phi \mathbb{E}_{\mathcal{T}_i \in \mathcal{S}} \log P(\boldsymbol{\theta}_i | f_\omega(e_i))$ and multiple steps of gradient descent are possible. $\omega$ is initialized by $\phi$.

The lower level optimization is on the likelihood of meta-training pre-trained models, and the upper level is to optimize the generalization on unseen pre-trained models. This

bi-level optimization ensures that the optimized meta initialization can be fast adapted to the unseen task with a few labeled data. We denote the upper level optimization as $\mathcal{F}(\phi)$. This bi-level optimization can be efficiently solved by first-order method, similar to first-order MAML (FO-MAML) Finn et al. [2017].

## 4.2 DISTRIBUTIONALLY ROBUST BLACK-BOX MODEL FUSION

The basic DFL2L Eq. (3) aims to improve the black-box fusion model generalization to unseen testing tasks. However, due to the underlying complex task distributions, a fixed number of pre-trained models have significant uncertainty and are insufficient to represent the actual task distributions. We believe this is because the number of pre-trained models is relatively smaller than that in the underlying task distribution (data-based meta-learning), which consists of a very large number of tasks (more than 100K tasks). The limited number of pre-trained models (e.g., 100 tasks) is highly insufficient to approximate the exact actual task distribution and contains very high uncertainty. Thus, there is a big gap between the task distribution that the pre-trained models represent and the underlying actual task distribution. That is to say, the task distribution uncertainty (pre-trained models represent) in the data-free setting is much more significant than in the data-based meta-learning setting. Uncertainty modeling is more necessary than data-based meta-learning. Furthermore, since the number of pre-trained models is relatively small, it is easy to overfit these models but cannot generalize to the unseen tasks.

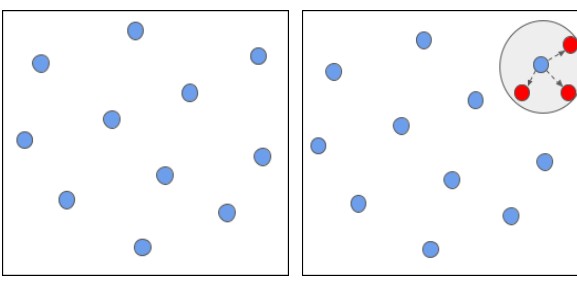

(a) Task embedding with basic DFL2L Eq. (3)  (b) DRO task embedding with Eq. 7

Figure 3: Illustration of task embedding with basic DFL2L Eq. (3) and DRO DFL2L Eq. 7. The blue dots denotes the task embedding $e$, and the red dots denote the perturbed embedding $e'$

To capture this uncertainty and encourage the black-box fusion model to generalize well to unseen testing tasks, we thus propose a DRO framework for encouraging the diversity of the task embeddings to capture the diversity in the underlying training task distributions. On the other hand, it has been shown that the DRO with $\mathcal{L}_2$ loss and Wasserstein ball constraint can be approximately formulated by an empirical risk minimization plus a regularization term. The regularization term is a dual norm of the model weights Kuhn et al. [2019]. That is to say, it can reduce overfitting and helps generalization with DRO. Suppose the raw task embedding $\{e_1, e_2, \cdots, e_N\}$ with distribution $\mu$. We use the ambiguity set, $\mathcal{P} = \{\nu | W(\mu, \nu) \leq \delta\}$, for characterizing the uncertainty of task embedding, where $\nu$ denotes the distribution of the perturbed task embeddings and $\delta$ is the constant threshold. $W(\mu, \nu)$ is the Wasserstein distance defined for a pair of probability distributions $\mu$ and $\nu$ as the following:

$$W_2(\mu, \nu) = \left( \min_{\pi \in \prod(\mu, \nu)} \int c(e, e') d\pi(e, e')) \right)^{1/2}, \quad (4)$$

where $\prod(\mu, \nu) = \{\pi | \pi(A \times \mathbb{R}^d) = \mu(A), \pi(\mathbb{R}^d \times B) = \nu(B)\}$ and $c(e, e') = ||e - e'||^2$. $\pi(e, e')$ is the joint probability measure with marginal measures equal to $\mu$ and $\nu$ respectively. Using Wasserstein distance allows the uncertainty distribution to cover task embeddings beyond the training set, thus enabling the black-box model to generalize to unseen test tasks. Thus, we propose the following optimization for distributionally robust model fusion:

$$\max_{\phi} \inf_{\nu \in \mathcal{P}} \mathcal{F}(\phi) \quad (5)$$

$$\text{s.t. } \mathcal{P} = \{\nu | W(\mu, \nu) \leq \delta\}, \quad (6)$$

where the inner $\inf$ optimization in Eq. (5) is to optimize the worst-case task embedding distribution, and Eq. (6) is to constrain the neighboring task embeddings distribution $\nu$ are within the Wasserstein ball of the original task embeddings distribution $\mu$. We use the ambiguity set of the Wasserstein ball to approximate the underlying uncertainty since this ambiguity set contains an infinite number of distributions and thus can encourage diverse task distributions that would cover more tasks and improve generalization. Compared to Eq. (3), Eq. (5) focuses on the worse-case performance within a Wasserstein uncertainty set. By Lagrangian duality, we can convert the above optimization to the following unconstrained optimization:

$$\max_{\phi} \inf_{\nu \in \mathcal{P}} \mathbb{E}_{\nu}[\mathcal{F}(\phi) + \gamma W(\mu, \nu)], \quad (7)$$

where $\gamma$ is the regularization weight. The above optimization problem can be further converted into the following equivalent form by using the equivalent optimization from [Blanchet and Murthy, 2017]:

$$\max_{\phi} \mathbb{E}_{\mu} \inf_{e'}[\log P(\theta | f_{\phi_*}(e')) + \gamma c(e, e')], \quad (8)$$

where $e'$ is the perturbed task embedding. We illustrate the difference between the basic and DRO DFL2L in Figure 3.

**Algorithm 1** Offline DRO Model Fusion for Meta Training.

1: **REQUIRE:** Given a collection of $N$ tasks consisting of $\mathcal{C} = \{(\mathcal{T}_1, \boldsymbol{\theta}_1), (\mathcal{T}_2, \boldsymbol{\theta}_2), \cdots, (\mathcal{T}_N, \boldsymbol{\theta}_N)\}$, where $\mathcal{T}_i$ is the task identifier and $\boldsymbol{\theta}_i$ are the pre-trained model parameters for task $i$; $E_N$ is the number of meta-training epochs; The number of inner-loop optimization steps is $K$; regularization weight is $\gamma$. Divide the pre-trained model set $\mathcal{C}$ into meta training $\mathcal{S}$ and meta-validation set $\mathcal{Q}$. Randomly initialize the black-box network parameters $\boldsymbol{\phi}$.
2: **for** $q = 1$ to $E_N$ **do**
3:    **for** $t = 1$ to $N$ **do**
4:       randomly sample a pre-trained model $(\mathcal{T}_i, \boldsymbol{\theta}_i)$ from meta-training set $\mathcal{S}$ and maximize the objective $\log P(\boldsymbol{\theta}_i | f_{\boldsymbol{\phi}}(e_i))$ with gradient ascent w.r.t. $\boldsymbol{\phi}$ by $K$ steps gradient ascent, obtains $\boldsymbol{\phi}_*$.
5:       randomly sample a pre-trained model $(\mathcal{T}_j, \boldsymbol{\theta}_j)$ from $\mathcal{Q}$ and minimize $\log P(\boldsymbol{\theta}_j | f_{\boldsymbol{\phi}_*}(e_j)) + \gamma c(e_j, e')$ w.r.t. $e'$ by $K$ steps gradient descent, obtains $e_*$.
6:       maximize the objective $\log P(\boldsymbol{\theta}_j | f_{\boldsymbol{\phi}_*}(e_*))$ with gradient ascent w.r.t. $\boldsymbol{\phi}$.
7:    **end for**
8: **end for**

**Offline DFL2F** The meta-learned model initialization is thus the black-box model output of the average embedding. We summarize our proposed black-box distributionally robust model fusion for meta training in Algorithm 1. In algorithm 1, line 5-7 is to alternately update the task embedding $e$ and $\boldsymbol{\phi}$ for multiple epochs.

**Algorithm 2** Online DRO Model Fusion for Meta Training.

1: **REQUIRE:** Given a sequence of $N$ pre-trained models, i.e., $(\mathcal{T}_1, \boldsymbol{\theta}_1), (\mathcal{T}_2, \boldsymbol{\theta}_2), \cdots, (\mathcal{T}_N, \boldsymbol{\theta}_N)$, where $\mathcal{T}_i$ is the task identifier and $\boldsymbol{\theta}_i$ are the pre-trained model parameters for task $i$; meta-validation pre-trained model set $\mathcal{Q}$; inner-loop optimization steps is $K$. Randomly initialize the black-box network parameters $\boldsymbol{\phi}$.
2: **for** $t = 1$ to $N$ **do**
3:    maximize the objective $\log P(\boldsymbol{\theta}_t | f_{\boldsymbol{\phi}}(e_t))$ with gradient ascent w.r.t. $\boldsymbol{\phi}$ by $K$ steps gradient ascent, obtains $\boldsymbol{\phi}_*$.
4:    randomly sample a pre-trained model $(\mathcal{T}_j, \boldsymbol{\theta}_j)$ from $\mathcal{Q}$ and minimize $\log P(\boldsymbol{\theta}_j | f_{\boldsymbol{\phi}_*}(e_j)) + \gamma c(e_j, e')$ w.r.t. $e'$ by $K$ steps gradient descent, obtains $e_*$.
5:    maximize the objective $\log P(\boldsymbol{\theta}_j | f_{\boldsymbol{\phi}_*}(e_*))$ with gradient ascent w.r.t. $\boldsymbol{\phi}$.
6: **end for**

**Extension to Online DFL2F** We extend the above algorithm to the online setting of DEL2L. In this setting, the pre-trained models are revealed one after the other, and the goal is to sequentially meta-learn on this sequence of pre-trained models. The model training only goes through the meta fusion process by a single pass. Each arrived pre-trained model will be sequentially meta-trained with Eq. (8),

but previous learned pre-trained models will not be available to use. The extension algorithm to the online learning setting is shown in Algorithm 2.

### 4.3 META TESTING

During meta testing, all the pre-trained models are fused by the following way:

$$e_{init} = \frac{1}{N} \sum_{i=1}^{i=N} e_i, \tag{9}$$

$$\boldsymbol{\theta}_{init} = f_{\boldsymbol{\phi}_{meta}}(e_{init}), \tag{10}$$

where $e_{init}$ is the average embedding of all the pre-trained models, and $\boldsymbol{\phi}_{meta}$ is the optimal solution to the Eq. (8). Given a test task $\mathcal{T}_i$ with a few labeled training examples $\mathcal{D}_i^{tr}$, $\mathcal{L}(\boldsymbol{\theta}_t, \mathcal{D}_i^{tr})$ is the loss function for network with parameters $\boldsymbol{\theta}_t$ and $\mathcal{D}_i^{tr}$. The learned model initialization for meta testing is shown in Algorithm 3. In the algorithm, we use the fused parameters $\boldsymbol{\theta}_{init}$ as initialization for new unseen tasks. Each unseen task is equipped with a few labeled training data for each class to fine-tune the fused model initialization by gradient descent.

**Algorithm 3** Black-box model Fusion for Meta Testing.

1: **REQUIRE:** Given a collection of $M$ tasks consisting of $\mathcal{Z} = \{(\mathcal{T}_1, \mathcal{D}_1^{tr}, \mathcal{D}_1^{test}), \cdots, (\mathcal{T}_M, \mathcal{D}_M^{tr}, \mathcal{D}_M^{test})\}$ for meta testing. $\mathcal{T}_i$ is the task identifier. $\mathcal{D}_i^{tr}$ and $\mathcal{D}_i^{test}$ are training and testing data for task $\mathcal{T}_i$ respectively. $\boldsymbol{\phi}_{meta}$ are the learned optimal black-box fusion model parameters; number of adaptation steps $S$ and learning rate $\alpha$
2: $e_{init} = \frac{1}{N} \sum_{i=1}^{i=N} e_i$
3: $\boldsymbol{\theta}_{init} = f_{\boldsymbol{\phi}_{meta}}(e_{init})$
4: **for** $i = 1$ to $M$ **do**
5:    $\boldsymbol{\theta}_0 = \boldsymbol{\theta}_{init}$
6:    **for** $t = 1$ to $S$ **do**
7:       $\boldsymbol{\theta}_t = \boldsymbol{\theta}_{t-1} - \alpha \nabla \mathcal{L}(\boldsymbol{\theta}_{t-1}, \mathcal{D}_i^{tr})$
8:    **end for**
9:    evaluate the model performance with network $H_{\boldsymbol{\theta}_S}$ on task-specific testing data $\mathcal{D}_i^{test}$.
10: **end for**

## 5 EXPERIMENTS

We perform extensive experiments on both synthetic data and image datasets to demonstrate the effectiveness of the proposed method. We perform both offline and online model fusion for real image datasets.

### 5.1 BASELINES

To show the effectiveness of the proposed methods, we construct various baseline methods and compare them in the following.

**Vanilla averaging (VA)**, which averages all the pre-trained model parameters.

**Finetuning**, which uses a few labeled data from each unseen task to finetune the randomly initialized model.

**MAML** [Finn et al., 2017], which meta trains all the tasks with their training and testing data together. This setting is completely different from ours, for which only a pre-trained model for each task is available and private data is not available for meta-training.

**Optimal transport averaging (OTA)** [Singh and Jaggi, 2020], which uses optimal transport to calculate the weighted average of all the pre-trained models. Their models mainly work on the simpler case that all pre-trained models solve the same task.

**Model fusion with Gaussian process (MFGP)** [Lam et al., 2021], which uses Gaussian process for model fusion.

We provide more detailed descriptions of baselines in Appendix 1.1

## 5.2 OFFLINE SYNTHETIC DATA

Suppose we have $N$ pre-trained models, where each one is a regression model to learn the sinusoid function $g(x) = a\sin(x + \beta) + b$, where $(a, \beta, b)$ denotes the magnitude, phase and vertical shift of the sine function. Where $a$ is sampled from the range [0.1, 5], $\beta$ is sampled from the range $[0, 2\pi]$ and $b$ is sampled from [0, 3]. The function domain is [-5, 5]. Note that our sampled sinusoid function has a vertical shift to increase the task diversity. This task construction is more difficult than existing works [Finn et al., 2017]. The identifier of each task is simply a number from $1, \cdots, N$. We keep 100 pre-trained models for meta-training and validation; another 100 unseen tasks for meta-testing. The task embedding for task $i$, i.e., $e_i$, is simply from the look-up table. Task embeddings are jointly learned with the black-box model parameters $\phi$. The goal of DFL2L is to learn an initialization for the unseen testing sinusoid functions. During meta-testing, each unseen testing sinusoid function is presented with a few labeled data points used to adapt the meta initialization for the specific unseen tasks. We evaluate the model performance by the mean squared error across all the unseen testing sinusoid functions.

**Implementation Details**. The pre-trained model for a sinusoid function network is a two-layer fully connected layer with 50 hidden units for each layer. The task embedding dimension for each task is 100. The black-box model is structured with the same number of layers as the pre-trained models. Each fully connected layer of the black-box model takes the task embedding as input. The output dimension is the same as the number of parameters for the corresponding pre-trained model layer. We use the SGD optimizer to learn the black-box model with a learning rate of 0.03 for

Table 1: Performance comparison to baselines on sinusoid function regression

| Algorithm | 5-shot | 10-shot |
|---|---|---|
| VA | $2.295 \pm 0.08$ | $0.682 \pm 0.06$ |
| Finetuning | $9.629 \pm 0.19$ | $1.765 \pm 0.08$ |
| MAML | $6.973 \pm 0.06$ | $1.303 \pm 0.05$ |
| OTA | $2.271 \pm 0.07$ | $0.667 \pm 0.07$ |
| MFGP | $2.325 \pm 0.05$ | $0.691 \pm 0.09$ |
| Ours | $\mathbf{2.235 \pm 0.05}$ | $\mathbf{0.625 \pm 0.03}$ |

three epochs. $K = 1$. Among all the pre-trained models, We use 80 % of them for meta-training and 20 % of them for meta-validation.

**Results**. Table 1 shows the model performance across different baselines. Figure 4 shows some visualization results of different sinusoid functions. Due to the more challenging problem setup of data-free model fusion and adding the vertical shift term in synthesizing sinusoid functions, simply finetuning or applying MAML does not perform well. Among all the compared methods, our method outperforms other baselines. Finetuning performs the worst due to the lack of good initialization, and only a few labeled data are available. VA and OTA are the second-best because they learned better initial parameters than MAML and finetuning.

## 5.3 OFFLINE META LEARNING

To evaluate the effectiveness of the proposed method on more challenging real image datasets, we perform experiments on CIFAR-FS [Bertinetto et al., 2019] and Mini-Imagenet [Vinyals et al., 2016]. These two datasets are commonly used meta-learning datasets, consisting of 100 classes. We split each dataset into meta-training, validation, and testing subsets, where all the subsets are non-overlapping following Vinyals et al. [2016], Bertinetto et al. [2019]. Standard meta-learning methods train on many different tasks, where each task has its training data and testing data. Different from existing works, the training and testing data associated with each task are not available during meta-training but only a pre-trained model that solves a $R$-way classification problem. We get each pre-trained model by standard supervised learning on its training data. Each classification task randomly samples $R$ classes from the meta-training subset and uses 60% labeled data of the corresponding class for training and 20%, 20% labeled data of the corresponding class labeled data for validation and testing, respectively. We repeat the process for 100 tasks. After getting 100 pre-trained models, we use them as meta-learning resources to learn an initialization for novel tasks with unseen classes. This problem scale (number of pre-trained models) is much larger than that of existing model fusion methods [Singh and Jaggi, 2020]. Each testing task is sampled from the

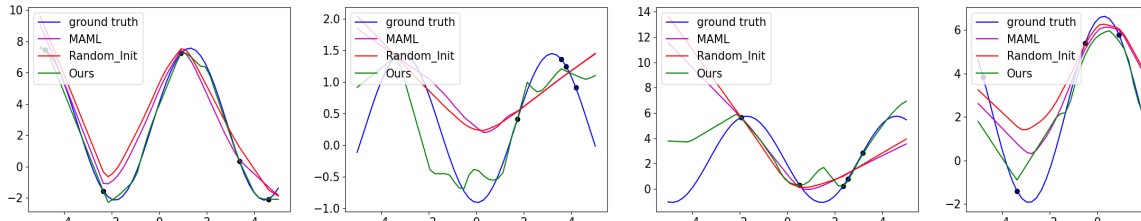

Figure 4: Visualization of different sinusoid functions with different adapttion method. We can observe that our method can better capture the sinusoid function structure. Due to the more challenging sinusoid function fitting than existing meta-learning setting, other compared methods cannot capture such wide range of function values.

meta-testing subset and has few labeled data to adapt the learned meta initialization to each unseen task. We evaluate the performance by the averaged accuracy and standard deviation over 600 unseen testing tasks that are sampled from the meta-testing subset.

**Implementation Details**. Each pre-trained model (network) is a four-layer convolution neural network, where each layer has 16 filters, similar to the standard meta-learning network [Finn et al., 2017]. The task embedding dimension for each task is 50. For the black-box model architecture, each layer is linear (fully connected) to predict the model parameters of the pre-trained model with task embedding as input. The output dimension of each linear layer is the number of parameters in the corresponding pre-trained model. We use the SGD optimizer to learn the black-box model with a learning rate of 0.03, $K = 1$. We perform each experiment for five runs and report each method's mean and standard deviation. Among all the pre-trained models, we use 80 % of them for meta-training and 20 % of them for meta-validation.

**Results.** Table 2 shows the evaluation results for 5-way classification on CIFAR-FS and Mini-Imagenet respectively. For CIFAR-FS, our method outperforms best baselines by $2.3\%$ and $3.7\%$ for 10-shot and 20-shot learning, respectively. For MiniImagenet, our method outperforms best baselines by $2.1\%$ and $3.1\%$ for 10-shot and 20-shot learning, respectively. The results show that simply finetuning the randomly initialized model parameters is insufficient to perform well. MAML performs better due to a better-learned meta prior to the model parameters. However, MAML does not significantly improve over simple finetuning due to the small number of training tasks. VA and OTA fuse model layerwise; they do not perform better because each pre-trained model trains on different tasks, thus lacking precise correspondence among different pre-trained models. MFGP also performs worse because GP cannot handle high dimensional network parameters and lacks meta-learning objectives. Our proposed method performs best because of the strong generalization ability of the black-box DRO fusion model with diversified task embedding.

Table 2: Compare to baselines in offline DFL2L on **CIFAR-FS** and **MiniImagenet** with 5-way and 10-way classification

|  | Algorithm | 10-shot | 20-shot |
|---|---|---|---|
| **CIFAR-FS** (5-way) | VA | $47.95 \pm 1.8$ | $50.88 \pm 1.6$ |
|  | Finetuning | $45.06 \pm 1.4$ | $48.81 \pm 1.9$ |
|  | MAML | $47.21 \pm 1.2$ | $50.35 \pm 1.7$ |
|  | OTA | $48.09 \pm 1.5$ | $51.16 \pm 1.5$ |
|  | MFGP | $47.68 \pm 1.6$ | $50.72 \pm 1.8$ |
|  | Ours | $\mathbf{50.42 \pm 1.5}$ | $\mathbf{54.86 \pm 1.2}$ |
| **MiniImagenet** (5-way) | VA | $35.07 \pm 1.7$ | $40.34 \pm 1.5$ |
|  | Finetuning | $30.25 \pm 1.8$ | $32.82 \pm 1.6$ |
|  | MAML | $34.51 \pm 1.2$ | $36.91 \pm 1.6$ |
|  | OTA | $35.25 \pm 1.6$ | $40.58 \pm 1.7$ |
|  | MFGP | $34.98 \pm 1.2$ | $40.06 \pm 1.4$ |
|  | Ours | $\mathbf{37.36 \pm 1.7}$ | $\mathbf{43.67 \pm 1.6}$ |
| **CIFAR-FS** (10-way) | VA | $34.73 \pm 1.7$ | $38.03 \pm 1.4$ |
|  | Finetuning | $27.86 \pm 1.4$ | $29.27 \pm 1.6$ |
|  | MAML | $31.97 \pm 1.9$ | $34.82 \pm 1.5$ |
|  | OTA | $35.05 \pm 1.6$ | $38.25 \pm 1.3$ |
|  | MFGP | $34.81 \pm 1.9$ | $38.17 \pm 1.7$ |
|  | Ours | $\mathbf{36.57 \pm 1.5}$ | $\mathbf{40.52 \pm 1.2}$ |
| **MiniImagenet** (10-way) | VA | $24.15 \pm 1.2$ | $28.06 \pm 1.5$ |
|  | Finetuning | $17.94 \pm 1.5$ | $18.40 \pm 1.6$ |
|  | MAML | $21.36 \pm 1.6$ | $25.87 \pm 1.8$ |
|  | OTA | $24.39 \pm 1.7$ | $28.42 \pm 1.9$ |
|  | MFGP | $22.78 \pm 1.4$ | $26.56 \pm 1.2$ |
|  | Ours | $\mathbf{27.57 \pm 1.9}$ | $\mathbf{32.68 \pm 1.7}$ |

**Effect of different number of training classes for each task** To evaluate the model performance difference where each task has more training classes, we experiment with the case that each task solves a 10-way classification problem. In this scenario, it is more difficult than the previous scenario. The results are shown in Table 2. For CIFAR-FS, our method outperforms best baselines by $1.5\%$ and $2.3\%$ for 10-shot and 20-shot learning, respectively. For MiniImagenet, our method outperforms best baselines by $3.2\%$ and $4.3\%$ for 10-shot and 20-shot learning, respectively. Compared to 5-way classification results, 10-way accuracy is lower due

to the more challenging problem. Ours outperforms all the baselines in both 5-way and 10-way settings.

**Hyperparameter Sensitivity and Ablation Study** We perform hyperparameter sensitivity analysis for $\gamma$ and ablation study for DRO in Appendix 1.2. Results show that our method is not very sensitive to hyperparameter variations, and the DRO regularization is effective.

## 5.4 ONLINE META LEARNING

This is different from Section 5.3, which focuses on the case that all the pre-trained models are available simultaneously. This section focuses on the setting that each pre-trained model sequentially arrives. In this case, when training on $i^{th}$ pre-trained model, all the previous $i-1$ pre-trained models are not available. In this case, most compared methods in offline DFL2L are not applicable. Hence, we only compare Finetuning and Online meta-learning (OML) [Finn et al., 2019a]. Note that OML does not follow the setting of DFL2L since it needs labeled raw data to train the model.

Table 3: Online DFL2L compare to baselines on **CIFAR-FS** and **MiniImagenet** with 5-way and 10-way classification

|  | Algorithm | 10-shot | 20-shot |
|---|---|---|---|
| **CIFAR-FS** **(5-way)** | Finetuning | $45.06 \pm 1.4$ | $48.81 \pm 1.9$ |
|  | OML | $46.91 \pm 1.5$ | $49.87 \pm 1.6$ |
|  | Ours | $\mathbf{49.18 \pm 1.2}$ | $\mathbf{53.06 \pm 1.7}$ |
| **MiniImagenet** **(5-way)** | Finetuning | $30.25 \pm 1.8$ | $32.82 \pm 1.6$ |
|  | OML | $33.83 \pm 1.2$ | $35.56 \pm 1.7$ |
|  | Ours | $\mathbf{35.89 \pm 1.1}$ | $\mathbf{41.93 \pm 1.8}$ |
| **CIFAR-FS** **(10-way)** | Finetuning | $27.86 \pm 1.4$ | $29.27 \pm 1.6$ |
|  | OML | $30.58 \pm 1.6$ | $33.57 \pm 1.8$ |
|  | Ours | $\mathbf{35.41 \pm 1.8}$ | $\mathbf{39.25 \pm 1.2}$ |
| **MiniImagenet** **(10-way)** | Finetuning | $17.94 \pm 1.5$ | $18.40 \pm 1.6$ |
|  | OML | $20.18 \pm 1.6$ | $24.31 \pm 1.8$ |
|  | Ours | $\mathbf{26.08 \pm 1.4}$ | $\mathbf{31.19 \pm 1.7}$ |

**Results** Table 3 shows the evaluation results for 5-way classification on CIFAR-FS and Mini-Imagenet respectively. For CIFAR-FS, our method outperforms best baselines by 2.3% and 3.2% for 10-shot and 20-shot learning, respectively. For MiniImagenet, our method outperforms best baselines by 2.1% and 6.3% for 10-shot and 20-shot learning, respectively. In addition, Table 3 shows the evaluation results for 10-way classification on CIFAR-FS and Mini-Imagenet respectively. For CIFAR-FS, our method outperforms best baselines by 4.8% and 5.7% for 10-shot and 20-shot learning, respectively. For MiniImagenet, our method outperforms best baselines by 5.9% and 6.8% for 10-shot and 20-shot learning, respectively. All the results show that our method substantially outperforms baselines by a large margin, demonstrating the effectiveness of the proposed DRO-based model fusion method. Compared to the offline DFL2L

setting, compared methods and our proposed methods perform relatively lower due to the more challenging nature of online DFL2L.

## 6 CONCLUSION

We propose a novel challenging meta-learning setting, i.e., Data-Free Learning To Learn, whose goal is to meta-learn a model initialization from several (sequential) pre-trained models without using their training data. The meta-learned initialization initializes new unseen tasks. To solve this challenging problem, we propose a Wasserstein distributionally robust optimization technique to fuse these existing pre-trained models into a single model without requiring training data, which is served as an initialization during the meta-testing stage. At last, extensive experiments in both offline and online settings demonstrate the possibility of our proposed DFL2F problem and the DRO-based model fusion solution's effectiveness. Future work includes extending the proposed method to more complex cases where each pre-trained model uses different architecture.

**Acknowledgements**

We thank all the anonymous reviewers for their insightful and thoughtful comments. This research was supported in part by NSF through grant IIS-1910492.

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
