# OpenReview forum: "Meta-Learning without Data via Wasserstein Distributionally-Robust Model Fusion"
_auai.org/UAI/2022/Conference — UAI 2022 Poster_

### Official Review · Reviewer_Ye5Q · 2022-04-12

**Q2(1) Originality/Novelty:** 4
**Q2(2) Significance/Impact:** 4
**Q2(3) Correctness/Technical Quality:** 4
**Q2(6) Clarity Of Writing:** 4
**Q6 Overall Score:** 8
**Q8 Confidence In Your Score:** 4

**Q1 Summary And Contributions:**

This paper proposes a new setting of data-free meta-learning and proposes a distributionally robust optimization framework.

**Q2 Assessment Of The Paper:**

More detailed information regarding each of these aspects is given below:

**Q2(4) Quality Of Experiments (Optional):**

4: Excellent: The experimental evaluation is comprehensive and the results are compelling.

**Q2(5) Reproducibility:**

3: Good: Key resources (e.g., proofs, code, data) are available and key details (e.g., proofs, experimental setup) are sufficiently well-described for competent researchers to confidently reproduce the main results.

**Q3 Main Strengths:**

1)This paper proposes a new setting of data-free meta-learning which is important for real-world applications.

2) For the data-free meta-learning tasks, this paper proposes a distributionally robust optimization framework to learn a black-box model to fuse and compress all the pre-trained models into a single network.

3) The proposed method is sufficient. The paper provides offline, online, and testing algorithms.

4) The proposed methods are evaluated on CIFARFS and MiniImagenet and achieve very good improvement.


**Q4 Main Weakness:**

No obvious weakness.

**Q5 Detailed Comments To The Authors:**

Please see Q3 for details.

**Q7 Justification For Your Score:**

I tend to accept this paper for its good performance and contribution.

**Q9 Complying With Reviewing Instructions:**

1: Yes.

---

### Official Review · Reviewer_YZu8 · 2022-04-13

**Q2(1) Originality/Novelty:** 4
**Q2(2) Significance/Impact:** 3
**Q2(3) Correctness/Technical Quality:** 3
**Q2(6) Clarity Of Writing:** 4
**Q6 Overall Score:** 7
**Q8 Confidence In Your Score:** 4

**Q1 Summary And Contributions:**

This paper proposes a new paradigm of meta learning and multi task learning, data free learning to learn (DFL2L), which learns a model that can be generalized to more unseen tasks from the pre-training model in multiple tasks. To solve this problem, the authors propose a black box model to learn a good model initialization, and propose a hard task-embedding mining module based on Wasserstein distribution robust optimization. Experiments shows the effectiveness of the proposed method.

**Q2 Assessment Of The Paper:**

More detailed information regarding each of these aspects is given below:

**Q2(4) Quality Of Experiments (Optional):**

2: Fair: The experimental evaluation is weak: important baselines are missing, or the results do not adequately support the main claims.

**Q2(5) Reproducibility:**

3: Good: Key resources (e.g., proofs, code, data) are available and key details (e.g., proofs, experimental setup) are sufficiently well-described for competent researchers to confidently reproduce the main results.

**Q3 Main Strengths:**

1. The ideas in this paper are novel and meaningful. DFL2L meta learns from pre-training models of multiple tasks, without using their data and labels. This is different from and more challenging than the general meta learning setting. Existing works pay little attention to this setting.  On the other hand, it is good that DFL2L inherits offline and online settings from task-based data meta learning.

2. The empirical study shows the advantages of the proposed method compared with other meta learning methods and model fusion methods. The results of regression and few-shot classification task show the effectiveness of the proposed method. In addition, the authors show ablation experiments on the hyperparameter tuning and the usage distributed robust optimization.

3. The paper is well written and all the figures are well illustrated.


**Q4 Main Weakness:**

1. The reason for using distributed robust optimization (DRO) framework needs to be further clarified.  We suggest adding an appropriate exploration of DRO application to classical meta-learning methods, or evidence that DRO is more suitable to DFL2L than from data-based meta learning.

2. The method is still a strong baseline worth reporting after removing the DRO module.  As an effective solution to a new problem, the method without DRO module deserves more empirical research.

3. The task embedding method is somewhat ambiguous.  For example, how to obtain the embedding vector from the task in a principled way? Some descriptions in the experiment are not clear either.

4. The usage of other baseline methods under DFL2L setting is not clear. Many methods for comparison are designed under different settings, so it is not trivial to adopt them under the setting of DFL2L.

5. More visualization results on model embedding and hard model-embedding mined by DRO module should be added.


**Q5 Detailed Comments To The Authors:**

Please refer to Q4

**Q7 Justification For Your Score:**

Novel idea effective experiments and good presentation.
Some technical points should be clarified and better illustrated.

**Q9 Complying With Reviewing Instructions:**

1: Yes.

---

### Official Review · Reviewer_fUzU · 2022-04-16

**Q2(1) Originality/Novelty:** 3
**Q2(2) Significance/Impact:** 2
**Q2(3) Correctness/Technical Quality:** 2
**Q2(6) Clarity Of Writing:** 3
**Q6 Overall Score:** 4
**Q8 Confidence In Your Score:** 3

**Q1 Summary And Contributions:**

This paper studies meta-learning problems when the data from the target tasks are not available. The main technique is to use a Wasserstein distributionally robust optimization technique to fine-tune these existing pre-trained models into a single model. This model is then served as an initialization during the meta-testing stage. Both online and offline settings are investigated and the proposed algorithm is tested on CIFAR and miniImagenet datasets.


**Q2 Assessment Of The Paper:**

More detailed information regarding each of these aspects is given below:

**Q2(4) Quality Of Experiments (Optional):**

2: Fair: The experimental evaluation is weak: important baselines are missing, or the results do not adequately support the main claims.

**Q2(5) Reproducibility:**

2: Fair: Key resources (e.g., proofs, code, data) are unavailable but key details (e.g., proof sketches, experimental setup) are sufficiently well-described for an expert to confidently reproduce the main results.

**Q3 Main Strengths:**

The idea of using DRO to fuse multiple pretrained models is novel.


**Q4 Main Weakness:**

My main concern is how DRO helps with the generalization and adaptation.

1. Specifying the constraints and setting the parameters is usually essential in DRO, which would affect how tight the adversarial player is constrained. The \delta parameter in the constraints seems to be missing in (7) and (8), which may be due to the primal dual transformation. But it is not intuitive that the \delta term does not affect the dual formulation at all.

2. In general, how DRO encourages the diversity is not clear. DRO, in theory, looks at the worst case. Whether the worst case indicates better performance depends on the constraints and inductive bias. But according to the experiments, the \gamma parameter does not affect the result a lot. This is kind of contradictory with the motivation of introducing DRO.


**Q5 Detailed Comments To The Authors:**

Since the main contribution is the application of DRO in meta learning, the paper probably should focus more on the intuition behind this idea and why it would work. The paper kind of assumes the readers to have knowledge about DRO and only refers to previous work about the details. But the details are essential to justification of the proposed method.

One important ablation study is probably varying different target task difficulties, compared to the source tasks. This would be useful to further test whether parameters in DRO makes a difference.


**Q7 Justification For Your Score:**

It’s interesting to see how DRO improves the meta-learning model fusion. However, there are some potential issues with the understanding of why and how the method works, as well as ablation studies in the experiments.


**Q9 Complying With Reviewing Instructions:**

1: Yes.

---

### Official Review · Reviewer_zyto · 2022-04-17

**Q2(1) Originality/Novelty:** 3
**Q2(2) Significance/Impact:** 3
**Q2(3) Correctness/Technical Quality:** 3
**Q2(6) Clarity Of Writing:** 3
**Q6 Overall Score:** 5
**Q8 Confidence In Your Score:** 3

**Q1 Summary And Contributions:**

This paper proposes a new setting for meta-learning: we don't have access to training data but we have pre-trained models and propose to fuse the models with task embeddings.

**Q2 Assessment Of The Paper:**

More detailed information regarding each of these aspects is given below:

**Q2(4) Quality Of Experiments (Optional):**

3: Good: The experimental evaluation is adequate, and the results convincingly support the main claims.

**Q2(5) Reproducibility:**

3: Good: Key resources (e.g., proofs, code, data) are available and key details (e.g., proofs, experimental setup) are sufficiently well-described for competent researchers to confidently reproduce the main results.

**Q3 Main Strengths:**

1. The new setting is potentially very useful since the computation cost of retraining a big model is big and how to reuse these published big models is an interesting problem.

2.  The results are better than the baseline methods on real and synthetic data.

3. The method to generalize the task embedding is interesting. The method learns to handle the worst-case of perturbed task embeddings.

**Q4 Main Weakness:**

1. The finetuning in section 5.1. According to my understanding, fine-tuning should be done on pre-trained models rather than randomly initialized models. Some clarifications are needed here. Authors may consider doing experiments with finetuning baseline as such: Best performance with single pretrained model and Average performance of all pretrained models.

2. The improvement brought by the proposed method is kind of minor, with 1.3% and 1.5%, which seems to suggest that the vanilla fusion model is enough.  Can you provide results on the synthetic dataset Without DRO?

**Q5 Detailed Comments To The Authors:**


1. The evaluation dataset. As far as I'm concerned, the proposed method needs lots of pre-trained models (since it needs to use distribution information, if the number of tasks is very small, the distribution information can be very weak), and can be difficult to use some published big models (e.g., GPT or BERT).  Do you have some possible solutions? Or do we need meta-learning here since the number of pre-trained models can be very small, e.g., 1 or 2?


2. The hyper-parameter of the ball. Selecting $\delta$ can be challenging, could you please provide more details?

**Q7 Justification For Your Score:**

The new setting is interesting but the improvement by the method is kind of minor.

**Q9 Complying With Reviewing Instructions:**

1: Yes.

---

### Decision · Program_Chairs · 2022-05-15

**Decision:**

Accept (Poster)

**Comment:**

Meta Review: DFL2L meta learns from pre-training models of multiple tasks, without using their data and labels. All reviewers think that the ideas in this paper are novel and meaningful. The setting is different from and more challenging than the general meta learning setting. Existing works pay little attention to this setting. On the other hand, it is good that DFL2L inherits offline and online settings from task-based data meta learning. R2 main concern is how DRO helps with the generalization and adaptation. The authors dispell the concerns of R2 through rebuttal and R2 has improved the score. I recommend acceptance.